# The Addition of Intravenous Propofol and Ketorolac to a Sevoflurane Anesthetic Lessens Emergence Agitation in Children Having Bilateral Myringotomy with Tympanostomy Tube Insertion: A Prospective Observational Study

**DOI:** 10.3390/children7080096

**Published:** 2020-08-15

**Authors:** Brandon d’Eon, Thomas Hackmann, A. Stuart Wright

**Affiliations:** 1Dalhousie Medical School, Dalhousie University, Halifax, NS B3H 4R2, Canada; brandon.deon@dal.ca; 2Department of Anaesthesia, Pain Management & Perioperative Medicine, Dalhousie University, Halifax, NS B3H 4R2, Canada; Thomas.Hackmann@Dal.Ca; 3Division of Pediatric Anaesthesia, IWK Health Centre, Halifax, NS B3K 6R8, Canada

**Keywords:** emergence agitation, sevoflurane, myringotomy, pediatric, propofol, ketorolac

## Abstract

The aim of this prospective observational study was to determine if children undergoing bilateral myringotomy and tympanostomy tube insertion with a sevoflurane anesthetic plus intravenous propofol and ketorolac experienced a lower incidence of emergence agitation than those receiving a sevoflurane anesthetic alone. Duration of procedure, length of stay in post-anaesthesia care and level of nursing effort required to care for patients were also assessed. In this study, 49 children younger than 13 years of age received a sevoflurane anesthetic. Fifty-one percent of these patients also received a single injection of propofol 1 mg/kg and ketorolac 0.5 mg/kg at the end of the procedure. Patients were assessed for emergence agitation using the Pediatric Anesthesia Emergence Delirium scale in the post-anaesthesia care unit. Four children receiving a sevoflurane anesthetic alone experienced emergence agitation, while no children receiving propofol and ketorolac experienced emergence agitation (*p* = 0.05). The length of stay until discharge from the hospital was 6.98 min longer for patients receiving propofol and ketorolac but did not reach statistical significance (*p* = 0.23). Nurses reported greater ease in caring for patients receiving the propofol and ketorolac injection (recovery questionnaire score 4.50 vs. 3.75, *p* = 0.002). In this study, adding a single injection of intravenous propofol and ketorolac to the end of a brief sevoflurane anesthetic for bilateral myringotomy with tube insertion was associated with a lower incidence of emergence agitation without significantly increasing the time to discharge from the hospital.

## 1. Introduction

Bilateral myringotomy with tympanostomy tube insertion (BMT) is among the most common day surgery procedures performed in the pediatric population. The most recent data from the United States indicates that approximately 699,000 children below the age of 15 undergo BMT each year [1]. Because of the brief duration of BMT, in many centres, the anesthetic is provided by inhalation of sevoflurane without establishing intravenous (IV) access. The reported incidence of emergence agitation (EA) among the children undergoing BMT with a sevoflurane anesthetic varies from 13% [2] to as high as 67% [3].

Emergence agitation manifests in the early recovery phase from general anaesthesia. It has been described as a state of agitation and confusion, with lack of awareness of surroundings [4]. Emergence agitation puts the child at risk of unintentional self-harm and can make the provision of post-operative care more challenging for health care providers [5]. The exact etiology of EA is unknown; however, risk factors include being a pre-school aged child, pre-operative anxiety, otorhinolaryngologic and ophthalmological surgeries, rapid emergence from anaesthesia, and the use of inhaled anesthetic agents such as sevoflurane and desflurane [5,6,7,8,9,10,11]. Post-operative pain is also suspected to contribute to emergence agitation, as several studies have reported decreased rates of emergence agitation following the administration of preoperative or intraoperative analgesics [12,13,14,15].

At our institution, two different approaches to delivering anaesthesia for BMT have evolved: (a) inhaled anesthetic with sevoflurane and nitrous oxide; or (b) the same anesthetic, at the end of which a single injection of propofol mixed with ketorolac is given using a butterfly canula. The IV propofol and ketorolac injection is added to prevent EA, as both agents have previously been described to reduce the incidence of EA when given with a sevoflurane anesthetic [12,15,16,17,18,19,20]. The aim of this prospective observational study was to determine if patients undergoing BMT with a brief sevoflurane anesthetic combined with IV propofol and ketorolac experienced a lower incidence of emergence agitation than those receiving the brief sevoflurane anesthetic alone. Secondary goals of the study were to determine length of stay in post-anesthetic care and the level of effort nurses reported while caring for patients.

## 2. Materials and Methods

Following approval by the IWK Health Centre Research Ethics Board on 13 June, 2016, a prospective observational study was conducted to examine the incidence of emergence agitation in pediatric patients following anaesthesia for bilateral myringotomy with tympanostomy tube insertion (Project # 1021229). Written informed consent was obtained from a parent or legal guardian prior to enrolling patients into the study and from nurses completing our recovery questionnaire prior to administration. Enrolment took place in the Day Surgery Unit at the IWK Health Centre (Halifax, NS, Canada) between 17 June, 2016 and 5 January, 2017.

Outpatients, younger than 13 years of age, scheduled for BMT were eligible to participate in the study. Excluded were patients who had a contraindication to receiving a volatile anesthetic, and those who had more extensive operations than BMT. Patients were also required to be English speaking, as the research team did not have the resources to administer the study in other languages. On the day of their scheduled operation, patients were given the opportunity to participate if they expressed interest in the study after having reviewed a pamphlet and if they met the inclusion/exclusion criteria. After written informed consent was obtained, patients underwent their scheduled BMT. Participation in the study did not impact the anesthetic provided. Whether the patient received the sevoflurane anesthetic alone or the sevoflurane anesthetic with the IV propofol and ketorolac injection was based solely on the clinical practice of the anesthesiologist.

The anesthetic technique delivered a mixture of sevoflurane and nitrous oxide via a facemask. Depending on the anesthesiologist’s practice, some of the patients received a dose of IV propofol 1 mg/kg mixed with ketorolac 0.5 mg/kg via a butterfly cannula (Terumo, Surshield™ Safety Winged Infusion Set, Vaughan, ON, Canada) at the end of the procedure. The canula was removed while in the operating room, and the puncture site was covered by an adhesive bandage. To blind the researcher to whether an injection was given, providers were instructed to place an adhesive bandage on all patients. All patients were recovered in the first-stage post-anaesthesia care unit (PACU 1), where they stayed until they were conscious, stable, and met discharge criteria. They were then reunited with their family in the second stage PACU (PACU 2) until they were deemed to be safe for discharge home.

Patients were assessed for the presence of emergence agitation using the Pediatric Anesthesia Emergence Delirium (PAED) scale following the protocol previously described by Sikich et al. [21]. Briefly, the researcher observed the patients for ten minutes upon emergence from their anesthetic and assigned them a score of 0–4 on each of the five items of the PAED scale. The sum of the values assigned to each item gave an overall score. The five items of the PAED scale are as follows: 1—the child makes eye contact with the caregiver, 2—the child’s actions are purposeful, 3—the child is aware of his/her surroundings, 4—the child is restless, and 5—the child is inconsolable.

Nurses who cared for the patients in PACU 1 were asked to report their observations using a Likert-type questionnaire created by the research team (Table 1). For safety reasons, nurses were not blinded to whether the patients had received propofol and ketorolac so that they could administer ibuprofen to those patients who had received sevoflurane alone and who were exhibiting signs of pain. The following information was collected after the PAED scale assessment was complete: age, sex, American Society of Anesthesiologists (ASA) physical status, preoperative pain management, behavior at time of induction, duration of procedure, PACU 1 length of stay, PACU 2 length of stay, PACU total length of stay, and total operation suite time.

The primary outcome was the occurrence of emergence agitation, which was defined as an overall PAED scale score greater or equal to 10. This value, established by Sikich et al., provides the best balance of sensitivity and specificity, 0.64 and 0.84, respectively [21]. The secondary outcomes were the duration of the operation, the length of the stay in PACU 1 and PACU 2, total length of time spent in post-anaesthesia care, total operation suite time, and level of nursing effort required to care for patients in PACU 1 as reported on our recovery questionnaire.

An in-house quality assurance project (unpublished data) suggested that the adjunct of propofol and ketorolac to sevoflurane anaesthesia for BMT markedly reduces the occurrence of emergence agitation. Assuming that this combination lowers the incidence to 5% compared to 38% of untreated children [12], we calculated a sample size of 46 patients to yield a statistically significant result with an alpha of 0.05 and a beta of 0.2 (power 80%). To allow for potential attrition of data, we aimed to study 50 patients.

IBM SPSS Statistics for Windows, version 25.0 (IBM Corp., Armonk, NY, USA), was used for statistical analysis. Categorical outcomes were reported as number and percentage and analyzed using Fisher’s Exact Test. Continuous outcomes were reported as mean and standard deviation and analyzed using Welch’s t-test. A *p* value ≤ 0.05 was considered statistically significant.

Prior to performing the analysis of the recovery questionnaire responses, the responses to questions 3–5 were reverse-coded to simplify the analysis. By reverse-coding questions 3–5, a higher value response to any question indicates that less effort was required to care for the patient. The six questions on the questionnaire were assessed for internal consistency using Cronbach’s alpha, and questions were removed if they did not correlate with the others. Responses to the remaining questions were used to calculate a single overall mean value for the questionnaire (recovery questionnaire score).

## 3. Results

A total of 51 patients having bilateral myringotomy with tympanotomy tube insertion were enrolled for assessment of emergence agitation following a sevoflurane anesthetic with or without adjunct IV propofol and ketorolac. Two patients were removed from the study after enrolment and excluded from the analysis. The first patient was removed because an additional procedure had been added to the scheduled BMT, and the second required a post-operative bedside procedure that interfered with the PAED scale assessment.

Of the 49 patients included in the analysis, 24 (49.0%) received a sevoflurane anesthetic and 25 (51.0%) received a sevoflurane anesthetic with IV propofol and ketorolac. There were no significant differences in age, sex, ASA physical status, preoperative pain management, or induction behavior between the two groups (Table 2). All patients who were treated preoperatively for pain control received acetaminophen except for one who also received ibuprofen. The latter patient belonged to the group that received only a sevoflurane anesthetic.

Emergence agitation was experienced by four patients in the study, all of whom had received only a sevoflurane anesthetic (Table 3). This result is statistically significant (*p* = 0.05). The duration of the operation and length of stay in PACU 1 was longer for patients receiving a sevoflurane anesthetic with IV propofol and ketorolac than those receiving a sevoflurane anesthetic alone (Table 3), although there was no significant difference in the length of stay in PACU 2 or the overall length of stay in hospital (Table 3).

When determining the internal consistency of the recovery questionnaire, Cronbach’s alpha was 0.66 when all six questions were included in the analysis and increased to 0.76 when question 3 was removed. Thus, question 3 was not included in the overall recovery questionnaire score calculation. There was a statistically significant difference in the recovery questionnaire score between the patients receiving a sevoflurane aesthetic and those receiving a sevoflurane anesthetic with IV propofol and ketorolac (Table 3).

## 4. Discussion

The results of this study indicate that adding a combination of IV propofol and ketorolac to the end of a brief sevoflurane anesthetic is associated with a decreased incidence of EA in children who undergo BMT. Our findings are consistent with prior studies showing the benefits of either propofol or ketorolac in reducing the incidence of EA [12,15,16,17,18,19,20]. Adding propofol 1 mg/kg to the end of a sevoflurane anesthetic for strabismus surgery, inguinal hernia repair, and magnetic resonance imaging, has been found to decrease the incidence of EA in children when compared to a saline control [16,17,18]. Intravenous ketorolac 1 mg/kg was also found to reduce the incidence of EA in 1—5 year-old children having BMT with a sevoflurane anesthetic when compared to a placebo control [12]. However, not all studies have found a benefit to the administration of end-of-procedure propofol or ketorolac with a sevoflurane anesthetic. Lee et al. performed a randomized control trial comparing IV propofol 1 mg/kg with a saline control for children undergoing adenotonsillectomies and found no difference in the incidence of EA [22]. Similarly, Kim et al. reported that IV ketorolac 1 mg/kg did not have an impact on the incidence or severity of EA in children undergoing elective surgery with a sevoflurane anesthetic [23].

Other agents that have been investigated for their potential to prevent EA in children having BMT with a sevoflurane anesthetic include fentanyl and dexmedetomidine. Intranasal fentanyl has been shown to decrease the incidence of EA in children after BMT when compared to a placebo [13,14]. The intranasal route of administration is a convenient choice, as it avoids the need for IV access and is easily administered intraoperatively. Unfortunately, fentanyl may lead to increased post-operative nausea and vomiting, which detracts from its use [14,19]. Several studies have found that dexmedetomidine also reduces EA and pain when given with sevoflurane anaesthesia [24,25,26]. In a randomized control study of children having an adenotonsillectomy with a sevoflurane anesthetic, dexmedetomidine was more effective at reducing EA than propofol when administered intravenously at the end of the procedure [27]. However, a retrospective chart review of children having BMT with sevoflurane anaesthesia found no difference in the in the PAED scale scores of children who received intranasal dexmedetomidine compared to those that did not [28].

A recent large retrospective cohort study of children, aged 9 months to 7 years, who underwent BMT under sevoflurane anaesthesia with the addition of intramuscular ketorolac, determined that normal middle-ear appearance at the time of surgery was paradoxically associated with elevated Face, Legs, Activity and Cry, Consolability (FLACC) scale scores potentially indicating significant post-operative pain [29]. In order to decrease the likelihood of this occurring, the authors suggested that children with normal middle-ear findings receive a combination of prophylactic ketorolac and fentanyl prior to their BMT [29]. However, as the authors acknowledged, there is overlap in the behaviors that indicate pain or agitation in younger children and the FLACC scale does not necessarily differentiate between these. It is possible that children in this study may have been experiencing emergence agitation instead of post-operative pain. In our opinion, adding propofol to IV ketorolac may offer a better solution to this problem since propofol’s hypnotic qualities and short action will address the emergence agitation phase of the recovery and intravenous ketorolac will provide faster onset of pain relief than the intramuscular route.

In our study, the duration of the operation was 2.1 min longer and the length of stay in the acute recovery phase, PACU 1, was 8.37 min longer for the patients receiving the propofol and ketorolac injection. These were anticipated findings, as the injection requires additional time to administer, and children receiving propofol at the end of a procedure would be expected to take longer to regain consciousness. Receiving a propofol and ketorolac injection, however, did not have a statistically significant impact on the length of time until discharge home. The mean length of stay in post-anaesthesia care for patients receiving the injection was less than 10 min greater than those receiving a sevoflurane anesthetic alone, which is consistent with the findings of a 2015 systematic review of patients receiving propofol at the end of a sevoflurane anesthetic to prevent EA [20].

From a nursing wellness and safety perspective, emergence agitation can be very difficult to manage and places nursing staff at risk of injury [5]. Nurses in our study reported significantly greater ease in caring for patients who had received propofol and ketorolac, compared to the group without the adjunct. Thus, this study demonstrates not only a beneficial effect for the patients but also shows greater satisfaction of the caregivers. We believe such measurements to be important for further investigations that report on perioperative care.

Although we attempted to avoid bias in this observational study by concealing the type of anesthetic received until after the PAED scale assessment was performed, the researcher was frequently unblinded. On several occasions, members of the child’s care team unintentionally revealed the type of anesthetic to the researcher prior to the PAED scale assessment being completed, or a bandage was not placed on the patient. Additionally, nursing staff had knowledge of the type of anesthetic provided prior to completing the recovery questionnaire. This may have impacted their responses.

In our study, a single injection of IV propofol and ketorolac added to the end of a sevoflurane anesthetic for BMT was associated with a decreased incidence of EA without significantly increasing the length of time until discharge home. This simple intervention may offer effective prophylaxis against emergence agitation in children undergoing bilateral myringotomy with tympanostomy tube insertion. Further evaluation with a randomized control trial may be warranted.

## Figures and Tables

**Table 1 children-07-00096-t001:** Halifax post-myringotomy anesthesia recovery questionnaire. Completed by nurses providing post-operative care to patients having bilateral myringotomy with tympanostomy tube insertion.

Please Read Each Statement below and Circle the Degree to Which You Agree or Disagree					
1. It was easy to rouse this child	1	2	3	4	5
2. The child was aware and responded appropriately	1	2	3	4	5
3. The child took a long time to wake up	1	2	3	4	5
4. The child was combative	1	2	3	4	5
5. I needed other staff to help me care for this child	1	2	3	4	5
6. I feel this child had the best recovery possible	1	2	3	4	5

1 = strongly disagree, 2 = disagree, 3 = neither agree nor disagree, 4 = agree, 5 = strongly agree.

**Table 2 children-07-00096-t002:** Characteristics of patients having bilateral myringotomy and tympanostomy tube insertion organized by type of anesthetic provided. Values are mean (SD) or number (proportion).

	Sevoflurane Anesthesia	Sevoflurane Anesthetic + IV Propofol and Ketorolac	*p*
	(*n* = 24)	(*n* = 25)	
Age; years	3.9 (2.4)	3.5 (2.1)	0.57
Sex; male	16 (66.7%)	12 (48.0%)	0.25
ASA physical status			0.32
ASA 1	16 (66.7%)	20 (80.0%)	
ASA 2	7 (29.2%)	4 (16.0%)	
Missing	1 (4.2%)	1 (4.0%)	
Pre-operative pain management	16 (66.7%)	14 (56.0%)	0.56
Induction behavior			0.59
Calm	16 (66.7%)	13 (52.0%)	
Crying	6 (25.0%)	6 (24.0%)	
Turbulent	2 (8.3%)	5 (20.0%)	
Missing	0 (0.0%)	1 (4.0%)	

ASA, American Society of Anesthesiologists; IV, intravenous.

**Table 3 children-07-00096-t003:** Patients experiencing emergence agitation after bilateral myringotomy and tympanostomy tube insertion and secondary outcomes categorized by type of anesthetic provided. Values are number (proportion) or mean (SD).

	Sevoflurane Anesthesia	Sevoflurane Anesthesia + IV Propofol and Ketorolac	*p*
	(*n* = 24)	(*n* = 25)	
Emergence agitation; PAED scale score ≥ 10	4 (16.7%)	0 (0.0%)	0.05
Duration of operation; minutes	5.0 (2.0)	7.1 (2.3)	0.001
PACU 1 length of stay; minutes	22.71 (9.82)	31.08 (12.97)	0.01
PACU 2 length of stay; minutes	39.83 (26.86)	36.32 (17.77)	0.59
PACU total length of stay; minutes	62.54 (24.94)	67.40 (14.73)	0.41
Total operation suite time; minutes	67.50 (24.68)	74.48 (14.78)	0.23
Recovery questionnaire score	3.75 (0.97)	4.50 (0.49)	0.002

IV, intravenous; PACU, post-anaesthesia care unit; PAED, pediatric anesthesia emergence delirium.

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
