# Peer review of "The Addition of Intravenous Propofol and Ketorolac to a Sevoflurane Anesthetic Lessens Emergence Agitation in Children Having Bilateral Myringotomy with Tympanostomy Tube Insertion: A Prospective Observational Study"

_children, 2020, doi:10.3390/children7080096_

Round 1

Reviewer 1 Report

This is simply an observational study, not randomized. Why could you not sign patients up to be randomized to which group, then have the anesthetics standardized, not just picked at the desire of the anesthesiologist?

You cannot compare your group of patients to ones who had more extensive surgeries. BMT not the same as eye or hernia surgery.

Move paragraph 2 p 6 where you talk of agents you did not use, routes you did not use to introduction, talking in general about things to help EA.

Nurses can be blinded and it is safe. Many studies safely blind participants.

Reviewer 2 Report

General comments
Overall, this is a small group prospective observational study for adjuvant add-on sevoflurane general anesthesia for tympanostomy tube insertion. Authors reported they added intravenous propofol and ketorolac into children having bilateral myringotomy for lessens emergence  agitation from general anesthesia, a well-known and common condition occurred. They concluded adding a single injection of intravenous propofol and ketorolac to the end of a brief  sevoflurane anesthesia for bilateral myringotomy with tube insertion was associated with a lower incidence of emergence agitation without significantly increasing the time to discharge from the hospital. The manuscript is concise, well written, and to the point.

I have some comments to challenge the authors to make this work even more attractive (I hope?)

  1. The primary outcome was defined as PAED score greater or equal to 10, why? How about 8 or 12?  
  2. How and Why do you choose and use the propofol dose as 1 mg.kg-1 and ketorolac in 0.5 mg.kg-1? Any preliminary data or references?
  3. Do use try propofol along without ketorolac and vise versa?
  4. How many anesthesiologist perform anesthesia in the study? Are they experienced? Are they resident? Did you analysis the performance different among them?
  5. How many nurses reported the questionnaire in PACU?
  6. Why the responses to questions 3, 4 and 5 should be reverse coded?
  7. Who do assesse patients were for the presence of emergence agitation using the Pediatric Anesthesia Emergence Delirium (PAED) ?
  8. About the randomization, did the study do any randomize?
  9. Table 3 and table 4 should be put together.

Reviewer 3 Report

To the Authors,

The issue is interesting and novelty enough. There are many issues need to clarify.

I suggest consult a statistician to confirm the the sample size calculation and p value evaluation, because the reduction from 38% to 5% is incredible and the authors described the P=0.05 is significant.

Please provide the total OR suite time, including Operation and total PACU stay times.

Round 2

Reviewer 1 Report

P 2 take out sentence in last paragraph about being anesthesiologists preference to do the propofol/ketorolac- this is a study, half got, half did not, this sentence is not needed.

Sample size good

p3. paragraph 4, how were patients randomized to what they got, please explain. Did they just let anesthesiologists do what they wanted and wait to get numbers?

How much more time is added in giving the med at end- paragraph 2 p 7

Reviewer 3 Report

To the Authors,

You had done a nice revision. I suggest accept this vision.

Best Regard,

Author Response

Dear Reviewer,

Thank you for taking the time to review our manuscript once again. We appreciate your recommendation to accept our manuscript for publication.

Sincerely,

Brandon d'Eon